# The Impact of Color Cues on Word Segmentation by L2 Chinese Readers: Evidence from Eye Movements

**DOI:** 10.3390/bs15070904

**Published:** 2025-07-03

**Authors:** Lin Li, Yaning Ji, Jingxin Wang, Kevin B. Paterson

**Affiliations:** 1Academy of Psychology and Behavior, Faculty of Psychology, Tianjin Normal University, Tianjin 300387, China; linpsy@outlook.com (L.L.); jiyaning0903@163.com (Y.J.); 2School of Psychology and Vision Sciences, University of Leicester, Leicester LE1 7RH, UK

**Keywords:** L2 readers, word segmentation, Chinese reading, color segmentation cues

## Abstract

Chinese lacks explicit word boundary markers, creating frequent temporary segmental ambiguities where character sequences permit multiple plausible lexical analyses. Skilled native (L1) Chinese readers resolve these ambiguities efficiently. However, mechanisms underlying word segmentation in second language (L2) Chinese reading remain poorly understood. Our study investigated: (1) whether L2 readers experience greater difficulty processing temporary segmental ambiguities compared to L1 readers, and (2) whether visual boundary cues can facilitate ambiguity resolution in L2 reading. We measured the eye movements of 102 skilled L1 and 60 high-proficiency L2 readers for sentences containing temporarily ambiguous three-character incremental words (e.g., “音乐剧” [musical]), where the initial two characters (“音乐” [music]) also form a valid word. Sentences were presented using either neutral mono-color displays providing no segmentation cues, or color-coded displays marking word boundaries. The color-coded displays employed either uniform coloring to promote resolution of the segmental ambiguity or contrasting colors for the two-character embedded word versus the final character to induce a segmental misanalysis. The L2 group read more slowly than the L1 group, employing a cautious character-by-character reading strategy. Both groups nevertheless appeared to process the segmental ambiguity effectively, suggesting shared segmentation strategies. The L1 readers showed little sensitivity to visual boundary cues, with little evidence that this influenced their ambiguity processing. By comparison, L2 readers showed greater sensitivity to these cues, with some indication that they affected ambiguity processing. The overall sentence-level effects of color coding word boundaries were nevertheless modest for both groups, suggesting little influence of visual boundary cues on overall reading fluency for either L1 or L2 readers.

## 1. Introduction

In our increasingly globalized world, second language (L2) acquisition has become an essential skill, with Mandarin Chinese emerging as one of the most strategically important languages to learn ([15]; [66]). Given the growing global demand for second-language learning, understanding cognitive mechanisms that enable learners to progress from novice to advanced proficiency is an increasingly important focus for research. A key theoretical debate in psycholinguistic research currently centers on whether advanced L2 readers employ the same cognitive strategies as native speakers or rely on distinct mechanisms for sentence processing and ambiguity resolution ([18]).

With the present research, we focus on an important challenge in learning to read Chinese as a second language that is unique to its writing system. Unlike other writing systems used worldwide, Chinese employs a character-based writing system that does not use spaces or other visual cues to demarcate word boundaries ([33]; [74]). This creates frequent temporary segmental ambiguities, where character sequences can be parsed to create alternative possible word analyses with distinct meanings ([79]). For skilled L1 Chinese readers, this segmental processing ambiguity does not appear to be a significant cause of reading difficulty. Instead, readers appear able to efficiently and accurately segment continuous unspaced character strings into words with ease ([5]; [32]; [31]; [30]; [73]), reading at approximately 250 words per minute, comparable to reading speeds for spaced alphabetic scripts like English ([35]). However, segmental ambiguities seem likely to challenge L2 Chinese readers, especially those who do not encounter similar segmental ambiguities in their native language ([2]). Understanding how L2 Chinese readers process and ultimately overcome segmental ambiguity is therefore crucial for developing effective pedagogical approaches that support the progression from novice to advanced L2 reading proficiency.

### 1.1. The Role of Visual Cues to Word Boundaries

Many writing systems, including alphabetic scripts like English, use interword spaces to visually mark word boundaries. A wealth of research shows that this low-spatial frequency cue plays an important role in alphabetic reading, helping readers to unambiguously identify words and guide their eye movements ([50], [51]; [30]). Moreover, reading is disrupted when the spaces between words are removed in such scripts, indicating that interword spaces play an important role in efficient reading performance ([39]; [48]; [53]; [43]; [64]; [47]; [44]). This raises the question of whether using spaces to mark word boundaries might improve reading efficiency in Chinese? For native Chinese readers, current research suggests that including interword spacing that correctly cue word boundaries can reduce processing time for ambiguous or complex sentences ([19], [20]). However, for more simple sentences, including interword spaces does not appear to significantly impact reading speed or comprehension accuracy, suggesting that interword spaces provide only limited benefits for skilled Chinese readers ([5]; [19], [20]; [30]; [73]; see also [21], and see [63], for studies investigating effects in Javanese—an unspaced syllabic script).

Fewer studies have investigated the potential benefits of interword spaces for L2 Chinese readers. Those studies that have addressed this issue nevertheless suggest that L2 readers can use these visual cues to guide segmentation decisions while also experiencing difficulty when interword spaces are used to incorrectly cue word segmentation ([56]; [11]). Interword spaces are also widely employed in Chinese language textbooks aimed at L2 beginner Chinese learners ([70]; [34]), suggesting that they may be useful in supporting L2 beginners. However, as visual boundary cues are absent in everyday Chinese text, L2 beginner Chinese learners may also experience difficulty when transitioning to authentic reading. Finally, some studies suggest that inserting spaces between words in Chinese text may disrupt normal reading efficiency by causing upcoming characters to be perceived at more eccentric visual locations than normal, thereby altering normal text layout and reducing acuity for upcoming characters ([51]; [4]; [44]).

Such observations have led researchers to explore alternative methods for marking word boundaries without altering standard text layout. One promising approach uses color to perceptually group characters belonging to the same word, thereby cueing word boundaries without introducing additional physical space (see [47]; [16]; [44] for research in alphabetic scripts; [45]; [78], [77]; [41], [42] for research in Chinese; and [63], for research in Javanese—an unspaced syllabic script). Studies in alphabetic reading suggest that alternating word color can compensate for the removal of spaces in scripts like Spanish that normally employ interword spaces, allowing readers to maintain native-like reading speeds and processing efficiency relative to normally spaced text ([44]; [47]). Benefits are also observed in morphologically complex alphabetic languages like Finnish, where using text color to demarcate morphemes appears to better support the segmentation of long, multisyllabic words compared to more conventional hyphenation methods ([16]).

Research in Chinese additionally suggests that skilled L1 reading can benefits from use of alternating text color to group characters into words. Several studies report that L1 reading is faster, with fewer and shorter fixations, when alternating text color marks word boundaries, taken as evidence that color cues are effective for helping skilled readers to recognize words ([45]; [78]; [41], [42]). However, less attention has been paid to the possible benefits for L2 Chinese readers. To our knowledge, only [79] ([79]) has investigated this issue to date, showing that L2 reading is faster when alternating text color correctly cues word boundaries. At the same time, using text color to incorrectly cue word boundaries has been shown to increase reading times by both L1 and L2 readers ([78], [79]). This suggests that both groups use color cues to process word boundaries even when these conflict with correct word segmentation. Studies to date have focused on relatively simple texts and have not directly investigated the influence of color cues on the processing of segmental ambiguities. Such effects may nevertheless be important for beginning readers, however, by helping them to more rapidly resolve segmental ambiguities. Given this absence of relevant research, we therefore addressed this question, by comparing the effects of color cues on the processing of temporary segmental ambiguities by high-proficiency L2 readers and skilled L1 readers.

### 1.2. Segmental Ambiguity in Chinese Reading

The present research focused on incremental segmental ambiguities, which are common in Chinese reading, occurring when a multi-character word contains an embedded shorter word. For instance, the three-character “incremental” word 体育馆 (meaning “stadium”) incorporates the two-character word 体育 (meaning “sport”) as its first two characters, creating temporary ambiguity between whole-word and embedded-word analyses of these characters. Current theoretical accounts diverge in their predictions about how this ambiguity is processed. Serial accounts ([46]) propose that readers initially segment text based on their knowledge that Chinese predominantly employs one- and two-character words. According to this approach, readers should initially identify a two-character embedded word within a longer “incremental” word before experiencing processing difficulty on encountering the residual third character. Readers should then attempt to revise this misanalysis by adopting a whole-word analysis of the ambiguity, with this reanalysis incurring a further processing cost.

By comparison, parallel accounts, including the Chinese Reading Model (CRM; [30]), propose that the alternative possible lexical analyses are considered in parallel ([23]; [31]). Within the CRM, it is assumed that on each fixation the reader attempts to segment a small set of characters within their perceptual span. While the perceptual span encompasses about one character to the left and two or three characters to the right of the fixated character for skilled college-aged L1 readers ([22]), it is likely to be smaller for L2 readers (see [13]; [24]). All possible character groupings within this region are assumed to become activated in parallel, with alternative segmental analyses competing for selection via mechanisms inspired by competition-based models of word recognition (e.g., [38]). Following this approach, when processing a multi-character sequence like 体育馆, lexical entries for both the whole word (体育馆) and its embedded word (体育) become activated and compete for selection. According to the CRM, the whole-word analysis usually wins this competition, as it receives greater bottom–up activation because it has more component characters. Parallel accounts of Chinese word segmentation therefore predict that skilled L1 Chinese readers prefer to adopt a whole-word analysis of an incremental segmental ambiguity.

[76] ([76]) compared these contrasting accounts of word segmentation in two eye movement experiments with skilled college-aged L1 Chinese readers. In Experiment 1, participants read sentences containing either an incremental target word (e.g., 酒精灯) or a control target word that was identical to the increment target’s embedded word (e.g., 酒精). The verb immediately preceding the target word differed across versions of the sentences so that it could combine either plausibly or implausibly with the target word in the control sentences (点燃酒精灯/酒精 [lit the alcohol lamp/alcohol] are both plausible, but 清洗酒精灯 [washed the alcohol lamp] is plausible, while 清洗酒精 [washed the alcohol] is implausible), and so also the embedded word in the incremental word sentences. However, in this experiment, the verbs always combined plausibly with the whole-word analysis of the segmental ambiguity.

Zhou and Li observed plausibility effects for the control target words but not the embedded words in the incremental sentences, which they took to show that readers initially adopted a whole-word analysis of the ambiguity without activating the embedded-word analysis. Experiment 2 independently manipulated the plausibility of the incremental words and their embedded words across different verb contexts. Plausibility effects were observed only for incremental target words and not the embedded words, which was taken as further evidence that readers preferentially adopted a whole-word analysis of the ambiguity. Findings from both experiments therefore were consistent with a parallel account exemplified by the CRM. These findings were subsequently replicated by [26] ([26]) for skilled college-aged L1 readers and older L1 readers, suggesting that segmental ambiguity processing is preserved in older age. What remains unclear, however, is whether L2 Chinese readers employ the same processing strategy as L1 readers.

Research on L2 ambiguity resolution has to date relied on offline methods to assess learners’ accuracy in identifying word boundaries ([58]; [57]; [72]), showing that L2 learners frequently make segmentation errors, adversely affecting their comprehension. However, L2 research has not investigated how these ambiguities are processed during reading in real time. The present research therefore addressed this issue, using eye movement measures to assess real-time ambiguity processing by skilled L1 readers and high-proficiency L2 readers, while also investigating the effects of visual boundary cues on this processing.

### 1.3. The Present Experiment

Similarly to [76] ([76]), the present research measured eye movements for sentences containing temporarily ambiguous three-character incremental words (e.g., “音乐剧” [musical]), where the initial two characters (“音乐” [music]) could also form a valid word. The embedded word was either plausible or implausible in the context of the immediately preceding verb, while the whole-word analysis was plausible regardless of the verb context. This enabled us to investigate whether L1 or L2 groups process the embedded word’s plausibility during reading. If both groups adopt a whole-word analysis of the ambiguity, in line with parallel processing accounts, no effects of embedded word ambiguity should be observed. However, if either the L1 or L2 group identify the embedded word, we would expect to observe a plausibility effect. Moreover, if this effect is observed in measures of initial ambiguity processing, this would suggest that they first selected this analysis of the ambiguity, potentially in line with a serial processing account.

We also examined the effects of color coding word boundaries. We did so by presenting the sentence stimuli under different color conditions: either in a standard mono-color format or with alternating text color used to demarcate word boundaries. Under one alternating text color condition, characters belonging to the whole-word analysis of an ambiguity were displayed in the same text color; under the other condition, the two characters belonging to the embedded word were displayed in the same text color while the third character was displayed in a different text color. This enabled us to test whether using alternating text color to mark word boundaries would influence decisions about how to segment the ambiguity. Presenting all three characters in the same color might aid selection of whole-word analysis, potentially facilitating processing of the ambiguity relative to the mono-color condition. Alternatively, presenting the embedded word’s characters in the same color, distinct from that of the third character, might encourage selection of the embedded-word analysis. In this case, we might observe a larger embedded word plausibility effect relative to the other two conditions. The manipulation would therefore reveal whether using alternating text color to mark word boundaries influences segmental ambiguity processing by L1 and L2 readers.

Following [76] ([76]), we additionally analyzed eye movements for the pre-target verbs that immediately preceded the target word. This allowed us to investigate parafoveal processing dynamics for the two groups of readers. Parafoveal processing refers to the acquisition of linguistic information from an upcoming word in the text (see [12]; [55]). This is seen as a hallmark of skilled reading, allowing readers to begin processing the following word before it is fixated. It was therefore of interest to establish whether skilled L1 and high-proficiency L2 readers exhibit similar parafoveal processing capabilities. A key question was whether either group could acquire parafoveal semantic information while fixating the pre-target verb. This question is highly controversial in research on alphabetic reading ([1]; [17]; [36]; [52], [54]), where it has generally been assumed that only lower-level orthographic and phonological information can be processed parafoveally, and not semantic information. However, there is growing evidence that such effects can be obtained in Chinese reading, potentially because of its more compact script ([69]; [71]; [40]). Moreover, this research suggests that Chinese readers can pre-process the contextual plausibility of parafoveal words. A key further question we therefore address in the present research is whether readers can process the plausibility of the embedded-word analysis of the segmental ambiguity while fixating the pre-target verb in our sentence stimuli, and whether such effects differ for L1 compared to L2 readers.

Demonstrating such effects would support the view that upcoming words can be segmented and lexically identified before being directly fixated (although note that [76] ([76]) observed no such effects in their study). Moreover, this analysis may also shed light on the parafoveal processing capabilities of high-proficiency L2 readers, and whether these differ from those of skilled L1 readers. Very little research to date has investigated L2 parafoveal processing, despite the importance of parafoveal processing in skilled readers. Existing research suggests that L2 readers exhibit parafoveal processing competence in alphabetic reading ([59]). However, Chinese-specific studies that have been conducted to date suggest that L2 parafoveal processing is limited in this script ([10]; [67]). We therefore conducted exploratory analyses comparing L1 and L2 parafoveal processing of segmental ambiguities, and whether this might be influenced by parafoveal visual boundary cues.

### 1.4. Summary

In summary, the present research investigates how color cues influence word segmentation in Chinese by both L1 (native) and L2 (non-native) readers. Specifically, it aims to directly test:Whether L1 and L2 readers use similar or different strategies to resolve segmental ambiguity during Chinese reading; andWhether highlighting word boundaries using alternating text colors can support ambiguity resolution for either group.

In addressing these questions, the research also offers broader insights into differences in L1 and L2 sentence processing. First, it contributes to understanding of eye movement patterns in L1 and L2 readers during Chinese reading, building on a relatively limited body of research. Second, it sheds light on the more general impact on eye movement behavior on using text color cues to mark word boundaries in text for both groups of readers. Finally, by analyzing eye movements just before readers encounter an ambiguous segment, the study offers new insights into the parafoveal processing abilities of L1 and L2 readers, and the role of parafoveal information in resolving segmental ambiguity.

## 2. Materials and Methods

### 2.1. Transparency and Openness

The present research was conducted in accordance with the principles of the Declaration of Helsinki ([65]) and received approval from the Ethics Committee at the Faculty of Psychology, Tianjin Normal University (protocol number: APB2024051604, for study entitled “The impact of color cues on word segmentation by L2 Chinese readers: Evidence from eye movements”).

Sentence stimuli, datasets, and analytic code for analyses conducted in R are publicly available via: https://figshare.com/s/d09c2b3cefc44a65f21c, 7 July 2025.

### 2.2. Participants

To determine the optimal sample size, we performed an a priori analysis in R using the simr package ([14]) based on pilot data from 28 participants (following [21]. A linear mixed-effects model analyzed pre-target region gaze duration, with effect sizes derived from model b-values ([8]). Power simulation revealed 82.3% power for 30 participants, 92% for 47 participants, and 95.2% for 58 participants (Figure 1).

To maximize the likelihood of observing any interaction effects, we recruited 102 L1 Chinese readers who were students aged 18–26 years (*M* = 21.6 years, *SD* = 1.9, 89 females) from Tianjin Normal University in Tianjin, China, and 60 L2 Chinese readers aged 18 to 33 years (*M* = 24.4 years, *SD* = 3.2, 46 females) from various universities in Tianjin. The L2 participants were native speakers of a diverse range of mostly alphabetic orthographies that varied in their use of spaces for marking word boundaries (see Table 1). All L2 participants had at least two years’ experience learning Chinese (*M* = 6 years, *SD* = 3.5), and 95% of them had achieved level 5 of the Hanyu Shuiping Kaoshi (HSK-5, the standard Chinese language proficiency test for non-native speakers administered by Ministry of Education of the People’s Republic of China, which comprises six levels), indicating that they had moderate- to high-level proficiency.

The L2 participants also completed the Language History Questionnaire, version 3 (LHQ 3.0, [28], which assesses second language proficiency ([75]). This questionnaire includes self-reported data on learning experiences, language usage, and proficiency (including overall performance and reading proficiency), collected via a web-based interface (LHQ 2.0, [27]). The L2 participants self-reported moderate to high levels of language proficiency overall, as well as specifically in reading skills (aggregated scores for L2 proficiency: M = 18, SD = 3; self-reported L2 reading proficiency: M = 4, SD = 0.92).

All participants had normal or corrected vision, and none reported any instances of color blindness or reading difficulties.

### 2.3. Stimuli and Design

We employed a 2 (participant group: L1, L2) × 2 (embedded word plausibility: plausible, implausible) × 3 (text color: mono-color, embedded word segmentation, incremental word segmentation) mixed experimental design. The stimuli were 90 pairs of sentences. Each pair included a three-character incremental word in which the first two characters could form an embedded word (see [76]). Additionally, each sentence included an interchangeable two-character verb that immediately preceded the incremental word and was used to manipulate embedded word plausibility. Under one condition (plausible embedded word), the verb (e.g., 欣赏, meaning “enjoy”) was followed by a three-character incremental word (e.g., 音乐剧, meaning “musical”) that was plausible in this context and in which its first two characters could form an embedded word (e.g., 音乐, meaning “music”) that also was plausible. In the second condition (implausible embedded word), the verb was changed (e.g., 观看, meaning “watch”) so that while the incremental word remained plausible, its corresponding embedded counterpart became implausible. The incremental words were consistently plausible regardless of the verb context; however, the plausibility of the embedded words varied depending on which verb was used. See Table 2 for an example stimulus set. The sentence stimuli ranged from 15 to 21 characters in length (M = 19.5 characters), with the target word always located near the middle of each sentence. Across the stimulus set, the pre-target verbs were matched for both word and character frequency, as well as number of character strokes (see Table 3). Previous research using these stimuli has demonstrated that sentences containing contextually plausible or implausible target words are perceived as equally acceptable, with no difference in cloze predictability for their incremental words ([26]).

Sentence stimuli were displayed in either mono-color or alternating text colors. In the mono-color condition, all characters were presented normally, as black text on a white background. In the alternating text color condition, text color was used to mark word boundaries. In one condition (incremental word segmentation), the three characters belonging to the incremental word were shown in the same color, while the preceding and following words were shown in a different color. In the other condition (embedded word segmentation), the first two characters in the incremental word that formed its embedded word were displayed in a uniform color, while the third character in the incremental word was presented in a different color. The preceding and following words were shown in alternating text colors. See Table 2 for an example of these sentence displays.

The sentence stimuli were allocated to six lists. Each began with four practice sentences, 90 experimental sentences, and 20 filler sentences (all of which were plausible). The experimental sentences comprised 15 sentences from each of the six experimental conditions. These were assigned to each list using a Latin square, ensuring that one version of each sentence appeared in each list and each list included an equal number of stimuli from each experimental condition. Each list’s experimental and filler sentences were presented in random order. An equal number of participants in both groups were assigned pseudo-randomly to each list.

### 2.4. Apparatus

An EyeLink 1000 Plus eye-tracker (SR Research, Ottawa, ON, Canada) recorded each participant’s right-eye gaze location every millisecond during binocular viewing. This system has high spatial (< 0.01° RMS) and temporal (1000 Hz) resolution. A head and chin rest was used to minimize head movement. Sentences were presented on a high-resolution 24 inch LCD screen (1920 × 1080 pixels, 150 Hz refresh rate) in Song font. Mono-color text was presented as black (RGB: 0, 0, 0) text on a white (RGB: 255, 255, 255) background. Alternating color text was presented as red (RGB:255, 0, 0) or blue (RGB: 0, 0, 255) on a white background. At a 75 cm viewing distance, each character subtended about 1° horizontally, so was of normal size for reading.

### 2.5. Procedure

Participants took part individually and received a detailed explanation of the experimental procedure prior to its start. This included instructing participants that some sentence presentations might be in different colors but that they should read all the sentences normally and for comprehension. At the start of the experiment, a 3-point horizontal calibration procedure was performed across the same line as each sentence was presented (ensuring 0.35° or better spatial accuracy for all participants). Calibration accuracy was checked before each trial, and the eye-tracker recalibrated as required to maintain high spatial accuracy throughout the experiment. At the start of each trial, a fixation cross equal in size to one character was presented on the left side of the screen. Once the participant fixated this location, a sentence was presented with its first character replacing the square. Participants pressed a response key once they finished reading each sentence. The sentence was then replaced on 25% of trials by a comprehension question requiring a yes or no response. Participants responded by pressing one of two response keys, with these responses recorded by the computer. The experiment lasted approximately 40 min per participant.

## 3. Results

Accuracy answering comprehension questions was high for all participants (>85%, L1, *M* = 92%, *SD* = 0.04; L2, *M* = 86%, *SD* = 0.07, *p* < 0.001), indicating that both groups understood the sentences well.

Following standard procedures, fixations shorter than 80 ms and longer than 1200 ms were removed (affecting 5.6% of fixations for L1 participants and 3.4% of fixations for L2 participants). The remaining data were analyzed by linear mixed-effects models (LMEMs; [3]) using R (version 4.3.2; [49]) and the lme4 package ([7]). *p*-values were estimated using the lmerTest package ([25]). For binomial variables, generalized LMEMs were conducted with the Laplace approximation. Continuous variables were log-transformed ([62]). A maximal random-effects structure was used ([6]). For sentence-level measures, participant group and text color were treated as fixed factors; in target word analyses, participant group, embedded word plausibility, text color, and their interaction served as fixed factors while participants and stimuli constituted crossed random effects. If model convergence issues arose, the random-effects structure was simplified by initially trimming it for stimuli—beginning with the removal of random effect correlations followed by random slopes ([37]). Contrasts were defined using sliding contrasts (the contr. sdif function) in the MASS package ([60]). Following convention, t/z values > 1.96 were considered significant.

We analyzed eye movements across the full sentence and at the word level for the two-character pre-target verbs and three-character incremental target words. The sentence-level analyses provided general insights into L1 versus L2 differences in eye movement behavior in reading, as well as the overall impact of text color coding segmental ambiguity (See Figure 2). For these analyses, we report sentence reading time (SRT, the time from the onset of a sentence display until the participant pressed a response key to indicate that they had finished reading the sentence), average fixation duration (AFD, the mean length of fixations), the overall number of fixations (NF), the number of regressions (NR, backward eye movements in the sentences), and average forward saccade amplitude (AFS, the mean length of forward-directed eye movements, which are reported as the number of characters traversed by a saccade).

Word-level analyses allowed us to directly examine L1–L2 differences in how text color cues influence the processing of the segmental ambiguity. The effects observed at the target region provided insight into how these factors impact foveal (fixational) processing of the ambiguous segment. By comparison, effects at the pre-target region revealed how parafoveal processing contributes to resolving segmental ambiguity before the ambiguous segment is directly fixated.

For the word-level analyses, we report measures informative about first-pass reading; that is, the initial processing of a word prior to a fixation to its right or a regression from that word. These measures comprised first-fixation duration (FFD; duration of the first progressive fixation on a word), single-fixation duration (SFD; first-pass fixation duration for words receiving only one first-pass fixation), and gaze duration (GD; sum of all first-pass fixations on a word). We also report word-level measures that are informative about later processing; namely, total reading time (TRT; sum of all fixations on a word) and regression-in (RI; probability of a regression to a word).

The word-level analyses enabled us to test hypotheses regarding the processing of segmental ambiguity and the effects of color coding for both participant groups. [61] ([61]) cautioned that using multiple dependent measures in eye movement research increases the risk of false positives, particularly through practices such as ‘cherry picking’ statistically significant results from individual measures. To mitigate this risk, we distinguished between hypothesis-driven analyses, grounded in prior research, and exploratory analyses that offer a broader descriptive account of the observed effects. Specifically, when examining plausibility effects related to the processing of the segmental ambiguity, we focused our hypothesis-driven analyses on the gaze duration and total reading time for the target word for both participant groups, as this measure would be sensitive to segmental processing during either the first-pass or later processing of the target region—noting that [76] ([76]) observed effects only in total reading time in their study. In addition to these hypothesis-driven analyses, we report exploratory analyses using a broader range of eye movement measures to investigate potential differences in how L1 and L2 readers process segmental ambiguities.

As noted above, effects at the pre-target region offered potential insights into the effects of color on parafoveal processing of the segmental ambiguity by L1 and L2 readers. If such effects were present, they were most likely to emerge in first-pass processing measures, such as gaze duration. We therefore focused hypothesis-driven analyses on gaze duration, while incorporating other eye movement measures in exploratory analyses to provide a more comprehensive account of potential L1–L2 differences. Additionally, [79] ([79]) reported L1–L2 differences in the effects of using alternating text color to mark word boundaries, with influences observed in both first-pass (gaze duration) and later (total reading time) measures. Accordingly, we focused our hypothesis-driven analyses on these two measures when assessing group differences in the effects of color coding. To date, no studies have directly examined the impact of text color coding on the processing of segmental ambiguity. Therefore, in the present experiment, we also analyzed gaze duration and total reading time to investigate whether marking word boundaries with color influences the resolution of segmental ambiguity.

### 3.1. Sentence-Level Analyses

Main effects of group: main effects of group were informative about general L1–L2 differences in eye movement behavior during reading. L2 readers exhibited longer sentence reading times compared to L1 readers (SRT, 7823 vs. 3126 ms), made more and longer fixations (NF, 27.93 vs. 12.37; AFD, 243 vs. 208 ms), shorter forward saccades (AFS, 0.55 vs. 3.47 chars), and more regressions (NR, 5.33 vs. 3.32). These effects indicated that the L2 group read about twice as slowly, making more than twice as many fixations that were around 20% longer. L2 readers also made more regressions to reread text, while making short forward saccades, characteristic of a cautious character-by-character reading strategy. By comparison, L1 readers appeared to process multiple characters on each fixation, traversing about 3–4 characters on each saccade, consistent with the perceptual span reported for skilled readers in previous research ([22]).

These sentence-level findings provided clear evidence of differences in the overall eye movement patterns of L1 and L2 readers during sentence reading. The findings were consistent with L2 readers adopting a more cautious, character-by-character reading strategy.

Table 4 shows the mean eye movements for sentence-level measures, while Table 5 summarizes the statistical effects.

### Effects of Color Coding on Sentence Processing

Sentence-level analyses also provided valuable insights into the overall effects of color coding for both L1 and L2 readers. In these analyses, we focused on total sentence reading time, as it offered the most comprehensive measure of how text color coding influences overall reading performance. This approach also helped to minimize the risk of false positives that can arise from selectively reporting results across multiple dependent variables ([61]). Numerically, the mono-color displays produced shorter reading times than either of the two color-coded displays for L1 readers but longer reading times compared to the color-coded displays for L2 readers.

This was not entirely borne out statistically, however. Statistically, we observed shorter reading times for mono-color compared to both embedded word segmentation (4857 vs. 4926 ms) and incremental word segmentation (4857 vs. 4815 ms) conditions, with an interaction for L1 vs. L2 groups only for the contrast between mono-color and incremental word segmentation (L1 effect = 122 ms, L2 effect = 321 ms). The absence of other interaction effects likely reflected high variance in sentence reading time for the L2 group, suggesting variability in the effects of color coding for these readers. The lack of an interaction for the contrast between incremental and embedded word segmentation suggested using text color to cue alternative analyses of the ambiguity did not differentially affect sentence reading time for the two groups.

Exploratory analyses of other sentence-level eye movement measures provided more detail about effects for L1 and L2 readers. These showed that, compared to the mono-color display, text color coding produced a small reduction in average fixation duration for both groups, as well as a small reduction in the number of fixations for the L1 group only. Statistically, these effects were characterized by slightly more and longer fixations for mono-color compared with the embedded word segmentation condition (AFD, 223 vs. 221; NF, 17.85 vs. 18.41) and a similar number of fixations that were longer on average for mono-color versus the incremental word segmentation condition (AFD, 223 vs. 220 ms). These were qualified by interaction effects (L1 vs. L2) for mono-color versus incremental word segmentation (AFD, L1 effect = 2 ms, L2 effect = 5 ms; NF, L1 effect = 0.58, L2 effect = 0.19). The absence of an interaction for the contrast between incremental and embedded word segmentation suggested that using text color to cue an alternative analysis of the ambiguity did not differentially affect the number or average duration of fixation by the two groups. Overall, the effects in these measures were small, suggesting that color coding the text had only weak effects in these variables.

Numerically, L2 readers made more regressions than L1 readers, with little variation across display conditions. Statistically, we observed fewer regressions for mono-color displays compared with either embedded word segmentation (3.92 vs. 4.19) or incremental word segmentation (3.92 vs. 4.13) displays. These effects were qualified by interaction (i.e., L1 vs. L2) effects. However, differences in regression rates were small across conditions, revealing a negligible effect of color-coding for either group.

Forward saccade length showed marked differences between L1 and L2 readers but little variation across the different display conditions. These were a little longer in mono-color vs. embedded word segmentation conditions (2.44 vs. 2.35 char) and mono-color vs. incremental word segmentation conditions (2.44 vs. 2.38 char), with no difference for embedded word vs. incremental word segmentation conditions (2.44 vs. 2.35 char). While these effects were qualified by interactions (i.e., L1 vs. L2) for mono-color vs. embedded word and embedded word vs. incremental word conditions, the variation in saccade length across display conditions was small, suggesting that text color coding had little influence on saccade-targeting.

Overall, we observed little effects of color coding on sentence-level eye movement behavior for L1 and L2 readers.

### 3.2. Word-Level Analyses

As already noted, the word-level analyses allowed us to directly test hypotheses concerning the effects of color coding on ambiguity processing by L1 and L2 readers. The analyses were also informative about the overall effects of text color coding on processes of word identification.

In this context, we focused on reporting the interaction effects associated with participant groups in the current findings, while other main effects and interactions are detailed in the Appendix A.

#### 3.2.1. Target Region

Table 6 shows the mean eye movements for the target region, while Table 7 summarizes the statistical effects.


*Effects of Group (L1 vs. L2) on Target Word Processing*


We first examined the group effects on target word processing. These were informative about overall L1–L2 differences in the processing of words during reading. Both first-pass fixation time (GD) and total reading time were overall much longer for L2 than L1 readers, with no difference in regression rates. Gaze duration and total reading time were both more than twice as long for L2 readers, suggesting that L2 readers experienced greater difficulty processing this region of text.


*Effects of Color Coding on Target Word Processing Across Groups*


We next examined the overall effects of color coding on L1–L2 processing. For L1 readers, we observed increased gaze duration and total reading time for target words when color coding cued an embedded-word analysis compared to the mono-color condition, and longer gaze duration and total reading time for target words when color coding cued an embedded-word analysis rather than a whole-word analysis. This suggested that using color to cue a dispreferred segmental analysis disrupted target word processing by L1 readers, in line with prior research ([56]; [11]; [79]).

By comparison, for L2 readers, we observed no overall effects of color coding on first-pass or later fixation time measures. This seemed likely to be a consequence of L2 readers having generally long fixation times for the target words regardless of text color coding (i.e., a ceiling effect).


*Target Word Plausibility Effects Across Groups*


Effects of target word plausibility at the target region allowed us to directly assess L1–L2 differences in segmental ambiguity processing across different color conditions. As a reminder, the embedded-word analysis of the ambiguous segment was either plausible or implausible depending on the preceding verb context, whereas the whole-word analysis was always plausible. Thus, observing a plausibility effect at the target word would indicate that readers had processed the embedded word’s interpretation. Crucially, if such an effect emerged during first-pass processing, it would suggest that readers initially selected an embedded-word analysis upon fixating the target word.

Our analysis revealed that L1 readers produced a plausibility effect in total reading time (444 vs. 455 ms), while L2 readers produced one in gaze duration (740 vs. 757 ms). These effects indicated that both groups showed sensitivity to target word plausibility and had processed the embedded-word analysis of the ambiguity. While this effect appeared in first-pass reading for L2 but not L1 readers, this seemed likely to be attributable to L2 readers spending much longer on the first-pass processing of this word compared to L1 readers, rather than them producing the effect earlier during processing.

Further analyses examined effects for each group for the different sentence displays. The L1 group produced plausibility effects in total reading time and regression-in for target words in mono-color displays, indicating that they were more likely to re-read these words when the embedded-word analysis was implausible rather than plausible. No such effects were observed when text color cued either an embedded or incremental word analysis of the ambiguity.

The L2 group showed no effect of plausibility in mono-color displays. However, they produced an effect in gaze duration for target words when text color cued an embedded-word analysis (734 vs. 776 ms). This suggested that the color coding prioritized an embedded-word analysis of the ambiguity. Crucially, however, the effect was due to longer fixation times for plausible than implausible embedded words. Note that this was opposite to the predicted effect and so would benefit from being replicated. One explanation may be that L2 readers experienced greater competition between embedded- and whole-word analyses during ambiguity processing when both were plausible as compared to when only the whole-word analysis was plausible.

The absence of a corresponding effect when text color cued a whole-word analysis (759 vs. 743 ms) suggested that L2 readers did not initially consider an embedded-word analysis when processing the target word in these sentences. For these sentences, we instead observed a plausibility effect in regressions back to the target word due to elevated regression rates when the embedded word analysis was plausible (28 vs. 32). One possibility is that L2 readers preferentially adopt a whole-word analysis of the ambiguity but were more likely to re-inspect the target word when an embedded-word analysis also was plausible. Effects of target word plausibility are illustrated in Figure 3.

#### 3.2.2. Pre-Target Region

Table 8 presents the mean eye movements data for the pre-target region, while Table 9 summarizes the statistical effects.

We next considered the effects at the pre-target word region, which were informative about L1–L2 parafoveal processing of the segmental ambiguity.


*Effects of Group (L1 vs. L2) on Pre-Target Word Processing*


We first examined group effects, which were informative about overall L1–L2 processing differences. As with findings for the target region, first-pass fixation time (FFD, GD) and total reading time were overall longer for L2 than L1 readers, with no group difference in regression rates. Similar to target region findings, gaze duration and total reading time were both more than twice as long for L2 readers, further indicating that they experienced considerable processing difficulty.


*Effects of Color Coding on Pre-Target Word Processing Across Groups*


We next examined overall effects of text color coding on L1 and L2 processing at the pre-target region. No effects of text color were observed at the pre-target region for L1 readers. By comparison, L2 readers produced longer total reading times for color-coded displays relative to mono-color displays, suggesting that L2 readers spent overall longer on this region when text color was used to cue word segmentation. We also observed a first-pass effect (in GD) for L2 readers when text color cued an embedded-word analysis of the ambiguity relative to the mono-color conditions. This suggested that L2 readers could process text color coding parafoveally and that this affected pre-target word processing when it cued a dispreferred analysis of the ambiguity.


*Word Plausibility Effects Across Groups*


Effects of target word plausibility at the pre-target region allowed us to directly assess L1–L2 differences in parafoveal processing segmental ambiguity across different color conditions. As already noted, the embedded-word analysis of the ambiguous segment was either plausible or implausible depending on the preceding verb context, whereas the whole-word analysis was always plausible. Thus, observing a plausibility effect at the pre-target region would indicate that readers had processed the embedded word’s interpretation of the ambiguity while fixating this earlier region. Crucially, if this effect emerged during first-pass processing, it would suggest that readers had accessed the embedded-word analysis of the ambiguity parafoveally.

For L1 readers, we observed first-pass (GD) and total reading time effects of embedded word plausibility (GD, 263 vs. 254 ms; TRT, 377 vs. 362 ms). The effect in GD indicated that L1 readers could parafoveally process the embedded word’s plausibility. For L2 readers, we observed embedded word plausibility effects in first-pass reading time (FFD, GD) as well as total reading time. These first-pass effects showed that L2 readers also could parafoveally process the embedded word’s plausibility (FFD, 269 vs. 261 ms; GD, 543 vs. 483 ms; TRT, 813 vs. 704 ms). The effect appeared to emerge earlier in processing for the L2 than L1 group, first appearing for the L2 group in the first fixation duration. However, this might have been a consequence of the L2 group having generally longer first fixation duration, so that effects emerged in this measure rather than in gaze duration.

Further analyses examined effects for each type of color display. For L1 readers, there was little effect of embedded word plausibility on processing at the pre-target region. Under the mono-color condition, we observed a plausibility effect in gaze duration (266 vs. 254 ms), suggesting that L1 readers processed the embedded word’s plausibility parafoveally. A similar effect in total reading time when text color cued an incremental analysis (379 vs. 357 ms) suggested that L1 readers engaged in greater re-reading of the pre-target region when the embedded word was implausible.

For L2 readers, we observed more interesting effects. For all three sentence types, we observed effects of embedded word plausibility in first-pass reading (gaze duration), due to longer fixation times at the pre-target region when this was followed by an implausible than plausible embedded word (mono-color, 544 vs. 492 ms; embedded word, 537 vs. 477 ms; incremental word, 547 vs. 480 ms). This indicated that L2 readers processed the embedded word’s plausibility parafoveally regardless of text color coding.

A similar effect was observed in total reading time (mono-color, 841 vs. 739 ms; embedded word segmentation, 788 vs. 679 ms; incremental word segmentation, 811 vs. 695 ms). This indicated that L2 readers engaged in re-reading of the pre-target region when the embedded word was implausible. Under the mono-color and embedded word conditions, the effects in regression-in provided further evidence for this effect of embedded word plausibility on the re-reading behavior of the L2 group (mono-color, 0.30 vs. 0.23; embedded word segmentation, 0.26 vs. 0.22).

L2 readers may have shown more extensive parafoveal processing effects at the pre-target region compared to L1 readers because of their much longer first-pass processing times at this region. Their longer processing time may have enabled them to engage in greater parafoveal processing compared to L1 readers. The effects of target word plausibility at the pre-target region are illustrated in Figure 4.

## 4. Discussion

The present research used eye movement measures to investigate the influence of visual boundary cues on segmental ambiguity processing by skilled native (L1) Chinese readers and high-proficiency second language (L2) readers. In total, 102 skilled L1 and 60 L2 readers read sentences containing a temporarily ambiguous three-character incremental words (e.g., “音乐剧” [musical]), where the initial two characters (“音乐” [music]) could also form a valid word. The embedded word was either plausible or implausible in the context of the immediately preceding verb (e.g., 观看/欣赏 [音乐]剧, watch/enjoy [music] musical), while the whole-word analysis was plausible regardless of the verb context. This enabled us to investigate whether both groups would access the plausibility of the embedded word while processing this ambiguity. This was important, as it would imply that readers processed the ambiguity in line with a serial-based account of segmental processing ([46]) rather than a parallel-based account like the Chinese Reading Model (CRM; [30]).

Sentence stimuli were presented using either a neutral mono-color display providing no word segmentation cues or color-coded displays marking word boundaries. Color-coded displays employed either (a) uniform coloring to promote a whole-word analysis of the segmental ambiguity or (b) contrasting colors for the two-character embedded word versus final character to promote an embedded-word analysis of the ambiguity and so induce a segmental misanalysis. This approach allowed us to test specific hypotheses concerning (1) L1–L2 processing segmental ambiguities during Chinese reading and (2) the use of color cues to support this ambiguity processing.

Additionally, the sentence- and word-level findings provide broader insights into L1–L2 reading behaviors. Moreover, by analyzing eye movements before readers fixated an ambiguous segment, the study provides new insights into the parafoveal processing abilities of L1 and L2 readers and its role in segmental ambiguity processing.

### 4.1. Sentence-Level Eye Movement Behavior

We began by examining overall sentence-level differences in eye movement patterns for the two groups. The L2 group read significantly more slowly than the L1 group—over twice as slow—by making more frequent and longer fixations, as well as more regressions. They also produced much shorter forward saccades, suggesting a cautious, word-by-word reading strategy. By contrast, the L1 group made forward eye movements that spanned 3–4 characters per saccade, consistent with processing approximately 4–5 characters per fixation. This aligns with estimates of the perceptual span for skilled readers ([22]). These patterns held across all sentence display types, indicating that L2 participants experienced greater reading difficulty, and adopted a more conservative reading strategy, even when word boundaries were clearly marked through text color coding. Despite these differences, both groups demonstrated high comprehension accuracy on follow-up questions presented after a subset of the sentence stimuli, suggesting that each group’s distinct reading strategy was effective in supporting sentence comprehension.

Next, we examined effects of using color cues to mark word boundaries on sentence-level eye movement behavior in both the L1 and L2 groups. Prior research suggested that color coding word boundaries can enhance reading efficiency for both L1 and L2 readers ([42]; [78], [79]), with similar findings reported for older adults and developing child readers ([41], [42]).

At first glance, our findings appear to contrast with these claims. Our sentence-level results showed minimal impact of alternating text color (used to indicate word boundaries) compared to mono-color presentations for both L1 and L2 readers. While some statistical effects emerged from the eye movement data, the mean differences were small and unlikely to reflect meaningful improvements in reading efficiency. However, a closer look at prior sentence-level findings in the literature suggests that there may be broader similarities between our results and prior research. Previous studies reported that using alternating colors to incorrectly mark word boundaries disrupted reading fluency for both L1 and L2 readers. By contrast, correctly coding word boundaries using alternating text color yielded only modest, and often minimal, facilitative effects. Thus, consistent with our findings, prior research suggests that color coding word boundaries offers limited benefit for skilled L1 readers or even for high-proficiency L2 readers. This is notable given that the L2 group in our study showed greater reading difficulty overall and relied on a conservative reading strategy to support adequate comprehension.

### 4.2. Target-Word Eye Movement Behavior

The specific focus of the present research was on investigating effects of text color cues on the processing of temporary segmental ambiguity. These ambiguities occur frequently in Chinese reading because of the absence of interword spaces or other visual cues to word boundaries. They are characterized by situations in which a sequence of characters is ambiguous between alternative lexical analyses, often where this ambiguity is subsequently resolved, and so disambiguated, by later text. We focused specifically on the processing of “incremental” segmental ambiguities that occur when a multi-character word contains an embedded shorter word. We gave as an example the three-character word 体育馆 (meaning “stadium”), which begins with the two-character word 体育 (meaning “sport”). During reading, this creates temporary ambiguity between a whole-word and an embedded-word analysis of these characters. We noted that current theoretical accounts diverge in their predictions about how this ambiguity is processed. Serial accounts (e.g., [46]) propose that readers initially segment the embedded word in strings before subsequently reanalyzing this misanalysis, which incurs an additional processing cost. By comparison, parallel accounts, like the Chinese Reading Model (CRM; [30]), propose that alternative analyses compete for selection, with the longer word usually winning. Crucially, we also noted that prior research has not investigated effects for L2 readers or the influence of text color cues on the processing of temporary segmental ambiguities. However, it seems that L2 readers, in particular, may benefit from visual cues that correctly identify word boundaries in ambiguous strings. This was therefore the focus of our study.

We first examined the effects of ambiguity by analyzing a target region containing the ambiguous segment. Within this region, characters were either presented in a neutral mono-color format or color-coded to cue either the embedded word or whole word’s interpretation of the ambiguity. Consistent with the sentence-level findings, L2 readers exhibited greater processing difficulty, as reflected in longer fixation times during both initial and later stages of processing. This highlighted that even high-proficiency L2 readers continued to face challenges in word-level processing.

A further notable finding was that L1 readers showed no difference in fixation time between the mono-color condition and the condition using incremental segmentation color coding. However, they exhibited a processing cost when the color coding cued an embedded-word segmentation, with longer gaze duration (GD) and total reading time compared to the mono-color condition. By contrast, no such effects were observed for L2 readers. These findings suggest that, at the word level, color coding that cues an incorrect interpretation of an ambiguity can disrupt processing for L1 readers. However, L2 readers did not show the same sensitivity to these segmentation cues.

Following the approach of [76] ([76]), we investigated the processing of the segmental ambiguity using verb contexts in which the embedded word’s interpretation was either contextually plausible or implausible, while the whole word’s interpretation remained consistently plausible. This allowed us to use plausibility effects in eye movement measures as an index of whether readers lexically accessed the embedded word during the processing of the ambiguity.

For the L1 group, the results revealed an embedded-word plausibility effect in total reading time on the target words, as well as in regressions back to those words, but only under the mono-color condition. Because these effects emerged in later processing measures, it suggested that L1 readers did not encounter segmental processing difficulty during first-pass reading of the target region. Furthermore, the absence of plausibility effects under the color-coded conditions indicated that alternating text color, whether supporting or miscuing segmentation, did not disrupt L1 readers’ processing of the ambiguity at the target word. Overall, the findings from the target region provide no evidence that color coding designed to miscue the segmentation of a segmental ambiguity adversely affected skilled L1 readers’ ambiguity processing.

For the L2 group, no embedded-word plausibility effects were observed under either the mono-color condition or when color coding cued a whole-word analysis of the ambiguity. This pattern suggested that, like L1 readers, L2 readers did not initially identify the embedded word in these contexts and thus may have employed a similar segmentation strategy. However, when color coding explicitly cued the embedded word’s interpretation, L2 readers exhibited a reversed plausibility effect: showing longer gaze duration for plausible embedded words than for implausible ones. This unexpected pattern may indicate that text color cues highlighting a plausible embedded word created a viable alternative to the incremental analysis, increasing competition between the two segmentation options and thereby extending the processing time for target words. By contrast, when the embedded word was implausible, it may have been more readily rejected, generating less competition and shorter gaze duration. Although these L2 findings are intriguing, they offer only limited evidence that color cues significantly influenced their processing of segmental ambiguities, while no such influence was observed for L1 readers.

### 4.3. Pre-Target Word Eye Movement Behavior

Following [76] ([76]), we also examined the effects at the pre-target verb to explore the possibility of parafoveal pre-processing of the ambiguity. One potential mechanism is that while fixating the verb, readers may access semantic information about the upcoming target word. This early access could enable them to begin evaluating the plausibility of different interpretations of the ambiguous segment before directly fixating on it. Importantly, such a process would constitute parafoveal processing, where linguistic information is extracted from a word in the peripheral vision and partially processed prior to direct fixation (for reviews, see [12]; [55]).

L1 readers showed an embedded-word plausibility effect in gaze duration under the mono-color condition and in total reading time under the incremental word color-coding condition. The effects in early measures such as gaze duration were particularly informative, as they reflected first-pass processing and may provide evidence for parafoveal processing of upcoming words. This finding suggested that, under the mono-color condition, L1 readers were able to parafoveally access the identity of the upcoming embedded word and evaluate its plausibility. This is a notable result, as it indicates that skilled L1 readers can extract semantic information from upcoming characters in the parafovea. We return to its implications shortly. By contrast, the plausibility effect observed in total reading time in the incremental word condition was less informative about parafoveal processing. Instead, it likely reflected later-stage re-reading behavior, whereby L1 readers spent more time revisiting the target word when the embedded word as plausible. Given our assumption that L1 readers initially adopt a whole-word analysis of the ambiguity, this may represent a post-hoc “sense check” to verify the initial interpretation. Nonetheless, the finding is still valuable, as it suggests L1 readers remain aware of alternative interpretations of the ambiguity, even if they do not act on them during initial processing.

L2 readers exhibited a broader pattern of effects at the pre-target verb. Across all three color conditions, they showed an embedded-word plausibility effect in gaze duration, indicating that they had processed the upcoming embedded word’s plausibility parafoveally. Notably, this effect was not modulated by the text color manipulation, suggesting that L2 readers could extract semantic information parafoveally from the upcoming embedded word regardless of the presence or type of segmentation cues. A parallel plausibility effect was observed in total reading time across all sentence conditions, with longer reading times at the pre-target verb when the embedded-word analysis was implausible. This further supports the conclusion that L2 readers actively evaluated the plausibility of potential segmentations while still fixating the verb, and that this evaluation influenced both early and later stages of their reading behavior.

Effects observed at the pre-target region provide evidence for parafoveal processing of word plausibility by both L1 and L2 readers; in this case, the plausibility of the embedded word analysis of the ambiguity. These findings suggest that both groups could extract complex semantic information regarding the contextual fit of an upcoming word in the parafovea. Such parafoveal semantic processing remains highly controversial in alphabetic reading (e.g., [1]; [17]; [36]; [52], [54]). However, there is growing evidence for its occurrence in Chinese reading (e.g., [69]; [71]; [40]). The parafoveal effects observed in the present study may therefore have important theoretical implications. First, they provide evidence that segmental processing of ambiguous words can begin at an early stage, prior to direct fixation on these words, during parafoveal preview. This suggests a level of semantic processing that precedes full lexical access during fixation. However, this interpretation must be considered alongside ongoing debates about whether parafoveal word segmentation in L1 Chinese reading also influences saccade-targeting behavior (see [29]; [68]).

Notably, [76] ([76]) did not report similar parafoveal processing effects in their study with L1 skilled readers. This discrepancy may be explained in one of two ways. First, the parafoveal effects we observed for L1 readers were relatively weak compared to those seen in L2 readers. We attributed this to L2 participants spending more time fixating the pre-target verb, allowing for more opportunity for parafoveal processing of upcoming words. It is possible that similar effects in Zhou and Li’s study were simply too subtle to be reliably detected. Second, the present study had substantially greater statistical power than Zhou and Li’s study, due to a larger participant sample and a larger stimulus set. This increased sensitivity may have led to detection of previously overlooked effects.

It is worth noting that the Chinese Reading Model (CRM) does not currently incorporate a mechanism for parafoveal word segmentation. However, the model could plausibly be extended to account for such processing. According to the CRM, word segmentation occurs within the perceptual span, which is generally assumed to apply to the currently fixated word. Yet it is reasonable to suggest that efficient reading may also involve segmentation of upcoming words within the parafovea. Our findings support this possibility, particularly for L2 readers, who appeared capable of evaluating the plausibility of the upcoming embedded word before direct fixation. One interpretation is that the L2 readers initiated word segmentation during parafoveal processing, with the result that the embedded word temporarily emerged as the preferred candidate in the competition for lexical selection during this early stage of processing.

These findings have important implications for ongoing debates about the development of parafoveal processing in L2 readers. While some studies suggest both beginning and proficient L2 readers are capable of sophisticated parafoveal processing ([59]), prior research in Chinese has generally reported more limited parafoveal processing among L2 readers ([10]; [67]). The present results contribute to this debate by providing evidence that high-proficiency L2 Chinese readers can engage in parafoveal processing, acquiring complex semantic preview information, such as the plausibility of upcoming words.

Finally, the present findings contribute to our understanding of the role of text color cues in supporting L2 word segmentation during reading. Previous research has suggested that color cues can facilitate word segmentation for both L1 and L2 Chinese readers ([79]), although such studies have not specifically examined their effects on the processing of segmental ambiguities. By contrast, the current findings indicate minimal benefit of text color cues for resolving segmental ambiguities. In fact, there was some evidence that color coding word boundaries may even slow reading for L1 readers.

This raises important questions about how the current results relate to prior research. One key difference lies in the design: previous studies manipulated color coding to cue correct or incorrect segmentation for all words in a sentence, whereas in the present study, all words except the segmental ambiguity were shown with correct segmentation, and only the ambiguous word was manipulated across conditions. Another important distinction is that prior studies typically reported only modest benefits of color coding when it cued correct segmentation, with larger effects observed when segmentation was miscued. Taken together, these observations suggest that the facilitative effects of text color may be more limited than previously claimed. Rather than broadly enhancing word segmentation, text color may exert its strongest influence when it disrupts segmentation, highlighting its potential to interfere more than to assist under certain conditions.

Clearly, further research is needed to deepen our understanding of these issues. Such studies could investigate a broader range of segmental ambiguities, including the overlapping word ambiguity examined by [9] ([9]), where character sequences such as 客运营 can be parsed either as a word formed by the first two characters (“passenger transport”) or by the last two characters (“operation”).

An important limitation of the present study is that we focused on high-proficiency L2 readers. Although this group demonstrated a cautious reading strategy, they may also have developed relatively strong word segmentation skills. It is possible that clearer effects of text color coding would emerge in less-skilled L2 readers, particularly those in the early stages of acquiring word segmentation strategies. Moreover, our L2 participant group was heterogeneous, comprising individuals with a broad range of first languages, including some that employ interword spacing (e.g., English) and others that do not (e.g., Thai or Japanese). This linguistic diversity may have introduced variability in segmentation strategies. Future research could explore L1-to-L2 transfer effects more directly by examining more homogeneous groups of L2 readers, specifically those whose first languages are either consistently spaced or unspaced. Such comparisons may help clarify how native language structures influence the acquisition and deployment of word segmentation strategies in L2 Chinese reading.

## 5. Conclusions

To conclude, our results, while complex, support several five key conclusions:

First, L2 readers adopted a slower, more effortful reading strategy than L1 readers yet achieved comparable comprehension, indicating that both groups used different but effective strategies for processing Chinese sentences.

Second, L1 readers processed segmental ambiguities efficiently and were only affected by misleading segmentation cues. By contrast, L2 readers showed slower, more effortful processing and may be more affected by lexical competition, indicating less automatic ambiguity resolution.

Third, L1 readers were selectively sensitive to color cues, showing disrupted processing when cues misaligned with correct segmentation. By contrast, L2 readers showed limited sensitivity to color cues, with minimal facilitation or disruption, suggesting reduced reliance on visual segmentation aids in resolving ambiguity.

Fourth, both L1 and L2 readers showed evidence of parafoveal processing of the embedded word’s plausibility, with stronger and more consistent effects for L2 readers, likely due to longer fixation times allowing greater semantic preview.

Crucially, these findings suggest that segmental processing can begin prior to direct fixation for both L1 and L2 readers, highlighting the role of parafoveal preview in initiating online processing of segmental ambiguity during reading.

## Figures and Tables

**Figure 1 behavsci-15-00904-f001:**
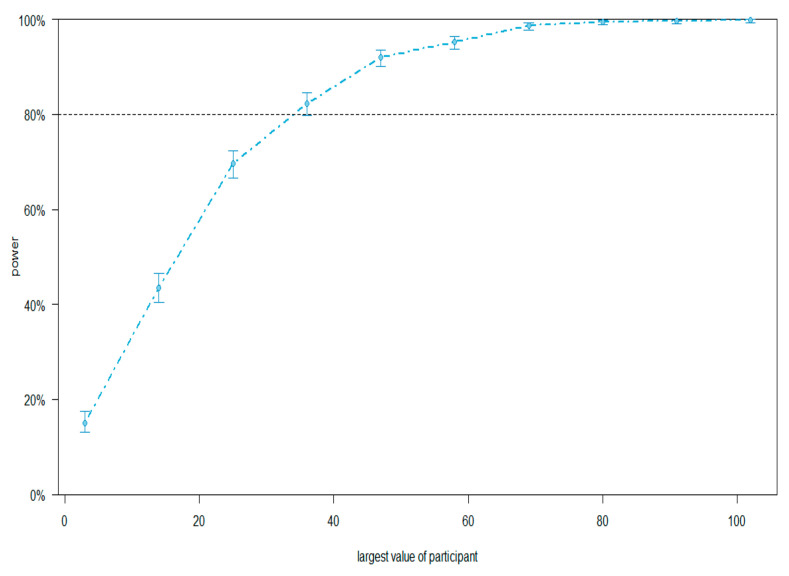
Power curve for experiment.

**Figure 2 behavsci-15-00904-f002:**
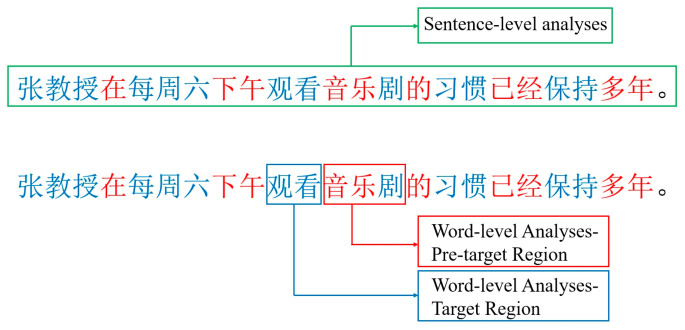
Example analysis regions. Note that this is shown in Chinese to illustrate the manipulation of text color. Sentence-level analyses correspond to content within the green box. The pre-target region in word-level analyses corresponds to content in the blue box. The target region in word-level analyses corresponds to content in the red box.

**Figure 3 behavsci-15-00904-f003:**
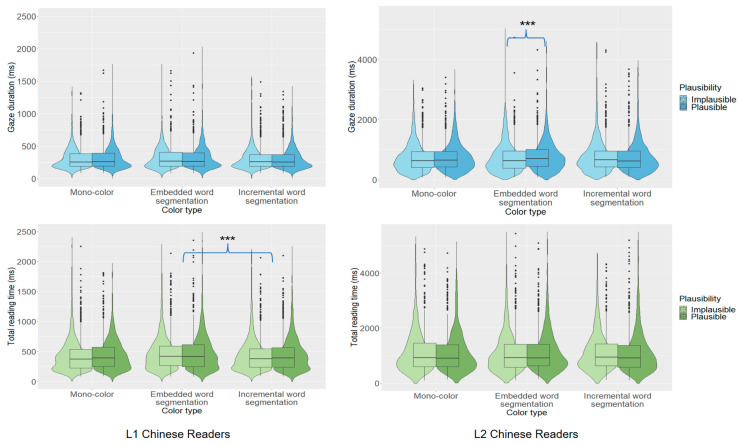
Word plausibility effects across groups for target region. Asterisks indicate statistically significant differences.

**Figure 4 behavsci-15-00904-f004:**
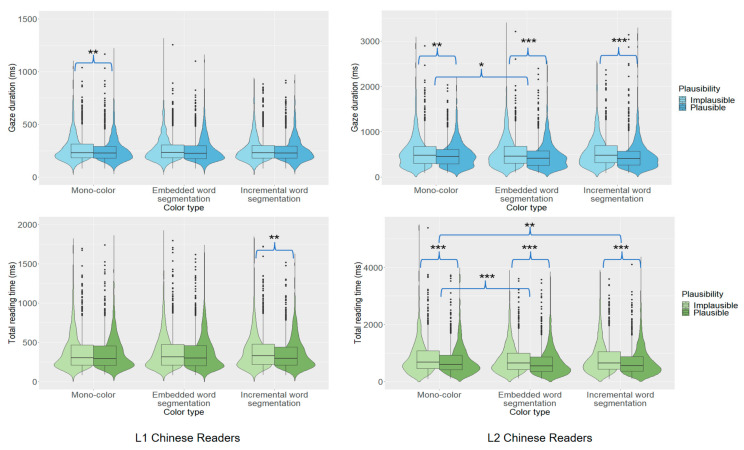
Word plausibility effects across groups for pre-target region. Asterisks indicate statistically significant differences.

**Table 1 behavsci-15-00904-t001:** Native language of the L2 participants in the experiment.

Native Language	Number of Participants	Script Type	Spaced	Word Boundary Cues
Arabic	1	Alphabetic	Y	Y
English	2	Alphabetic	Y	Y
Indonesian	1	Alphabetic	Y	Y
Khmer	4	Alphabetic	N	N
Lao	1	Alphabetic	Y	Y
Malay	3	Alphabetic	Y	Y
Mongolian	1	Alphabetic	Y	Y
Persian	1	Alphabetic	Y	Y
Tajik	1	Alphabetic	Y	Y
Thai	9	Alphabetic	N	N
Urdu	1	Alphabetic	N	Y
Uzbek	1	Alphabetic	Y	Y
Vietnamese	34	Alphabetic	Y	N

Note. The native languages of most of the L2 readers employ alphabetic scripts. However, some employ alphabetic scripts that use spaces to mark syllables rather than word boundaries (e.g., Lao, Thai, Vietnamese). One participant was a Mongolian speaker, which employs a spaced alphabetic script traditionally written vertically, while Arabic and Urdu speakers use spaced alphabetic scripts written from right to left.

**Table 2 behavsci-15-00904-t002:** Example sentence stimuli.

Embedded Word Plausibility	Color Type	Example Sentence Stimulus
Plausible	Mono-color	张教授在每周六下午*欣赏***音乐**剧的习惯已经保持多年. Prof. Zhang’s habit of *enjoying* the **music**al every Saturday afternoon has been maintained for many years.
	Embedded word segmentation	张教授在每周六下午欣赏音乐剧的习惯已经保持多年.
	Incremental word segmentation	张教授在每周六下午欣赏音乐剧的习惯已经保持多年.
Implausible	Mono-color	张教授在每周六下午*观看***音乐**剧的习惯已经保持多年. Prof. Zhang’s habit of *watching* the **music**al every Saturday afternoon has been maintained for many years.
	Embedded word segmentation	张教授在每周六下午观看音乐剧的习惯已经保持多年.
	Incremental word segmentation	张教授在每周六下午观看音乐剧的习惯已经保持多年.

Note. The pre-target verbs are shown in italics, the incremental words are underlined, and the embedded words are shown in bold. Note that all words were displayed as normal in the experiment, without italics, underlining, or bold. In the embedded word segmentation condition, the pre-target verbs and third character of the incremental words were shown as blue text (RGB: 0, 0, 255) while the embedded words were shown as red text (RGB: 255, 0, 0). In the incremental word segmentation condition, the pre-target verbs were shown in blue, and the incremental words were shown in red. Note that all words in the embedded word segmentation and incremental word segmentation conditions were displayed with alternating colors in the experiment, as shown in Table 2.

**Table 3 behavsci-15-00904-t003:** Characteristics of the pre-target verbs.

	Embedded Word Plausibility			
Characteristic	Plausible	Implausible	*t*	*p*
Word frequency	39 (42)	33 (48)	0.83	0.4
First character frequency	524 (603)	497 (670)	0.28	0.78
Second character frequency	623 (949)	782 (1242)	0.96	0.33
First character stroke	7.8 (2.7)	8.2 (2.4)	1.04	0.3
Second character stroke	7.8 (2.9)	7.9 (2.6)	0.3	0.77

Note. Word and character frequencies are reported in occurrences per million. The standard deviation of the mean is shown in parentheses. Note that the verbs served to manipulate the plausibility of the embedded word. The incremental words were always plausible.

**Table 4 behavsci-15-00904-t004:** Mean eye movement measures for sentence-level.

Measure	L1			L2		
	Mono-Color	Alternating Color		Mono-Color	Alternating Color	
		Embedded Word Segmentation	Incremental Word Segmentation		Embedded Word Segmentation	Incremental Word Segmentation
Sentence reading time (ms)	3000 (24)	3255 (58)	3122 (26)	8012 (176)	7764 (87)	7691 (85)
Average fixation duration (ms)	210 (1)	208 (1)	208 (1)	246 (1)	243 (1)	241 (1)
Number of fixations	11.87 (0.08)	12.80 (0.09)	12.45 (0.09)	28.02 (0.30)	27.95 (0.30)	27.82 (0.29)
Number of Regressions	3.13 (0.04)	3.49 (0.05)	3.34 (0.05)	5.23 (0.10)	5.32 (0.09)	5.45 (0.10)
Average forward saccade amplitude (characters)	3.55 (0.02)	3.40 (0.02)	3.46 (0.02)	0.54 (0.00)	0.56 (0.01)	0.55 (0.00)

Note. Standard errors are given in parentheses.

**Table 5 behavsci-15-00904-t005:** Summary of sentence-level statistical effects.

Contrast		Sentence Reading Time (ms)	Average Fixation Duration (ms)	Number of Fixations	Number of Regressions	Average Forward Saccade Amplitude (char)
Group	b	−0.90	−0.16	−15.55	−1.98	2.92
	SE	0.05	0.02	0.95	0.28	0.13
	t/z	−17.06 *	−9.14 *	−16.40 *	−7.08 *	22.76 *
Mono-color vs. Embedded word segmentation	b	0.03	−0.01	0.43	0.27	−0.07
	SE	0.01	0.00	0.12	0.05	0.01
	t/z	4.66 *	−4.64 *	3.49 *	5.10 *	−5.14 *
Mono-color vs. Incremental word segmentation	b	−0.01	0.01	−0.19	−0.26	0.04
	SE	0.01	0.00	0.12	0.05	0.01
	t/z	−1.68	6.07 *	−1.57	−4.86 *	3.16 *
Incremental word segmentation vs. embedded word segmentation	b	−0.02	−0.00	−0.24	−0.01	0.03
	SE	0.01	0.00	0.12	0.05	0.01
	t/z	−2.98 *	−1.43	−1.93	−0.25	1.98 *
Group × Mono-color vs. Embedded word segmentation	b	0.07	0.00	1.00	0.25	−0.17
	SE	0.01	0.00	0.25	0.11	0.03
	t/z	5.91 *	0.42	4.06 *	2.31 *	−6.60 *
Group × Mono-color vs. Incremental word segmentation	b	−0.06	−0.01	−0.78	0.02	0.11
	SE	0.01	0.00	0.25	0.11	0.03
	t/z	−4.71 *	−2.18 *	−3.14 *	0.15	4.03 *
Group × Incremental word segmentation vs. Embedded word segmentation	b	−0.01	0.01	−0.23	−0.26	0.07
	SE	0.01	0.00	0.25	0.11	0.03
	t/z	−1.19	1.75	−0.92	−2.48 *	2.58 *

Note. Asterisks indicate statistically significant effects, *p* < 0.05. Converged model for AFD, NF, SRT, AFSA, NR: depvar.lmer = lmer(depvar ~ COLOR * GROUP + (1|participant) + (1|item), datafile).

**Table 6 behavsci-15-00904-t006:** Mean eye movement measures for target region.

Measure	Target Region					
	Mono-Color		Alternating Color			
	Plausible Embedded Word	Implausible Embedded Word	Embedded Word Segmentation		Incremental Word Segmentation	
			Plausible Embedded Word	Implausible Embedded Word	Plausible Embedded Word	Implausible Embedded Word
L1 Chinese readers						
First fixation duration	229 (2)	225 (2)	227 (2)	229 (2)	224 (2)	223 (2)
Gaze duration	309 (4)	302 (4)	314 (4)	316 (5)	298 (4)	298 (4)
Total reading time	441 (7)	426 (7)	473 (8)	470 (7)	450 (7)	434 (7)
Regression-in (%)	30 (1)	27 (1)	29 (1)	31 (1)	30 (1)	29 (1)
L2 Chinese readers						
First fixation duration	267 (4)	264 (4)	266 (4)	265 (4)	265 (4)	267 (4)
Gaze duration	750 (16)	726 (16)	776 (16)	734 (17)	743 (17)	759 (17)
Total reading time	1082 (24)	1134 (26)	1125 (26)	1105 (25)	1112 (26)	1123 (24)
Regression-in (%)	29 (2)	30 (2)	32 (2)	30 (2)	32 (2)	28 (2)

Note. Fixation time measures are shown in ms. Standard errors are given in parentheses.

**Table 7 behavsci-15-00904-t007:** Summary of word-level statistical effects for target region.

Group	Contrast		Statistic	FFD	GD	TRT	RI
Group (L1 vs. L2)			b	−0.14	−0.81	−0.87	−0.07
			SE	0.02	0.04	0.05	0.12
			t/z	−7.46 *	−22.26 *	−18.20 *	−0.59
			p	<0.001	<0.001	<0.001	0.558
L1 Readers	Embedded word plausibility		b	−0.01	−0.00	−0.02	−0.08
			SE	0.01	0.01	0.01	0.05
			t/z	−0.95	−0.32	−2.27 *	−1.46
			p	0.341	0.747	0.023	0.146
L1 Readers	Mono-color vs. embedded word segmentation		b	−0.00	−0.03	−0.08	−0.10
			SE	0.01	0.01	0.01	0.06
			t/z	−0.49	−2.37 *	−5.84 *	−1.62
			p	0.627	0.018	<0.001	0.104
	Mono-color vs. Incremental word segmentation		b	0.02	0.02	−0.01	−0.06
			SE	0.01	0.01	0.01	0.06
			t/z	1.91	1.77	−0.94	−1.04
			p	0.056	0.077	0.345	0.301
	Embedded word segmentation vs. Incremental word segmentation		b	0.02	0.05	0.07	0.04
			SE	0.01	0.01	0.01	0.06
			t/z	2.42 *	4.18 *	4.92 *	0.59
			p	0.016	<0.001	<0.001	0.555
L1 Readers	Mono-color	Embedded word plausibility	b	−0.02	−0.02	−0.05	−0.19
			SE	0.01	0.02	0.02	0.09
			t/z	−1.39	−1.13	−2.40 *	−2.12 *
			p	0.164	0.258	0.017	0.034
	Embedded word segmentation	Embedded word plausibility	b	−0.00	0.00	−0.00	0.06
			SE	0.01	0.02	0.02	0.09
			t/z	−0.06	0.14	−0.15	0.66
			p	0.949	0.889	0.882	0.512
	Incremental word segmentation	Embedded word plausibility	b	−0.00	0.01	−0.03	−0.10
			SE	0.01	0.02	0.02	0.09
			t/z	−0.18	0.45	−1.37	−1.09
			p	0.859	0.656	0.172	0.275
L2 Readers	Embedded word plausibility		b	−0.01	−0.04	0.01	−0.11
			SE	0.01	0.02	0.01	0.07
			t/z	−0.84	−2.50 *	1.01	−1.56
			p	0.400	0.012	0.310	0.118
L2 Readers	Mono-color vs. embedded word segmentation		b	0.00	−0.01	−0.01	−0.09
			SE	0.01	0.02	0.02	0.08
			t/z	0.40	−0.72	−0.41	−1.17
			p	0.690	0.472	0.681	0.244
	Mono-color vs. Incremental word segmentation		b	−0.00	−0.01	−0.01	−0.02
			SE	0.01	0.02	0.02	0.08
			t/z	−0.27	−0.35	−0.62	−0.26
			p	0.791	0.723	0.535	0.798
	Embedded word segmentation vs. Incremental word segmentation		b	−0.01	0.01	−0.00	0.07
			SE	0.01	0.02	0.02	0.08
			t/z	−0.66	0.37	−0.21	0.91
			p	0.506	0.715	0.833	0.363
L2 Readers	Mono-color	Embedded word plausibility	b	−0.02	−0.05	0.04	0.01
			SE	0.02	0.03	0.02	0.11
			t/z	−1.19	−1.82	1.73	0.11
			p	0.234	0.068	0.084	0.911
	Embedded word segmentation	Embedded word plausibility	b	−0.01	−0.09	−0.03	−0.12
			SE	0.02	0.03	0.02	0.11
			t/z	−0.55	−3.43 *	−1.18	−1.05
			p	0.581	0.001	0.239	0.296
	Incremental word segmentation	Embedded word plausibility	b	0.00	0.02	0.03	−0.22
			SE	0.02	0.03	0.02	0.11
			t/z	0.29	0.92	1.20	−1.97 *
			p	0.774	0.358	0.230	0.049

Note. Asterisks indicate statistically significant effects, *p* < 0.05. Converged model for FFD, epvar.lmer = lmer(depvar ~ PI * COLOR * GROUP + (1 + PI|participant) + (1|item), datafile). Converged model for GD, TRT: depvar.lmer = lmer(depvar ~ PI * COLOR * GROUP + (1|participant) + (1|item), datafile). Converged model for RI: depvar.glmer = glmer(depvar ~ PI * COLOR * GROUP + (1|participant) + (1|item), datafile, family = binomial).

**Table 8 behavsci-15-00904-t008:** Mean eye movement measures for pre-target regions.

Measure	Pre-Target Region					
	Mono-Color		Alternating Color			
	Plausible Embedded Word	Implausible Embedded Word	Embedded Word Segmentation		Incremental Word Segmentation	
			Plausible Embedded Word	Implausible Embedded Word	Plausible Embedded Word	Implausible Embedded Word
L1 Chinese readers						
First fixation duration	229 (2)	235 (3)	226 (2)	229 (2)	229 (2)	233 (3)
Gaze duration	254 (3)	266 (4)	255 (3)	262 (3)	253 (3)	262 (4)
Total reading time	358 (6)	372 (7)	367 (6)	378 (7)	357 (6)	379 (6)
Regression-in (%)	24 (1)	24 (1)	23 (1)	27 (1)	26 (1)	28 (1)
L2 Chinese readers						
First fixation duration	264 (3)	273 (4)	261 (4)	267 (4)	259 (4)	266 (4)
Gaze duration	492 (9)	544 (12)	477 (11)	537 (11)	480 (11)	547 (11)
Total reading time	739 (17)	841 (20)	679 (16)	788 (17)	695 (17)	811 (18)
Regression-in (%)	23 (1)	30 (2)	22 (1)	26 (1)	26 (1)	28 (2)

Note. Fixation time measures are shown in ms. Standard errors are given in parentheses.

**Table 9 behavsci-15-00904-t009:** Summary of word-level statistical effects for pre-target region.

Group	Contrast		Statistic	FFD	GD	TRT	RI
Group			b	−0.13	−0.61	−0.66	−0.00
			SE	0.02	0.03	0.04	0.12
			t/z	−6.57 *	−20.13 *	−15.84 *	−0.03
			p	<0.001	<0.001	<0.001	0.98
L1 Readers	Embedded word plausibility		b	0.01	0.03	0.04	0.09
			SE	0.01	0.01	0.01	0.06
			t/z	1.90	3.40 *	3.43 *	1.54
			p	0.058	0.001	0.001	0.125
L1 Readers	Mono-color vs. embedded word segmentation		b	0.02	0.00	−0.02	−0.05
			SE	0.01	0.01	0.01	0.07
			t/z	1.75	0.39	−1.19	−0.67
			p	0.080	0.694	0.236	0.502
	Mono-color vs. Incremental word segmentation		b	0.01	0.01	−0.01	−0.18
			SE	0.01	0.01	0.01	0.07
			t/z	0.91	1.07	−0.84	−2.54 *
			p	0.364	0.285	0.404	0.011
	Embedded word segmentation vs. Incremental word segmentation		b	−0.01	0.01	0.00	−0.13
			SE	0.01	0.01	0.01	0.07
			t/z	−0.85	0.68	0.35	−1.90
			p	0.394	0.494	0.723	0.057
L1 Readers	Mono-color	Embedded word plausibility	b	0.02	0.04	0.03	−0.04
			SE	0.01	0.02	0.02	0.10
			t/z	1.58	2.72 *	1.41	−0.36
			p	0.115	0.007	0.158	0.720
	Embedded word segmentation	Embedded word plausibility	b	0.01	0.02	0.03	0.19
			SE	0.01	0.02	0.02	0.10
			t/z	0.78	1.62	1.56	1.91
			p	0.433	0.106	0.119	0.056
	Incremental word segmentation	Embedded word plausibility	b	0.01	0.02	0.06	0.11
			SE	0.01	0.02	0.02	0.10
			t/z	0.92	1.54	2.84 *	1.16
			p	0.359	0.124	0.005	0.245
L2 Readers	Embedded word plausibility		b	0.02	0.11	0.15	0.27
			SE	0.01	0.01	0.01	0.07
			t/z	2.52 *	7.91 *	10.26 *	3.85 *
			p	0.012	<0.001	<0.001	<0.001
L2 Readers	Mono-color vs. embedded word segmentation		b	0.02	0.03	0.07	0.11
			SE	0.01	0.02	0.02	0.08
			t/z	1.82	1.98 *	3.87 *	1.41
			p	0.069	0.048	<0.001	0.158
	Mono-color vs. Incremental word segmentation		b	0.03	0.02	0.05	−0.04
			SE	0.01	0.02	0.02	0.08
			t/z	2.30 *	1.07	2.85 *	−0.53
			p	0.021	0.284	0.004	0.598
	Embedded word segmentation vs. Incremental word segmentation		b	0.01	−0.02	−0.02	−0.16
			SE	0.01	0.02	0.02	0.08
			t/z	0.48	−0.91	−1.03	−1.94
			p	0.630	0.365	0.304	0.052
L2 Readers	Mono-color	Embedded word plausibility	b	0.02	0.07	0.12	0.43
			SE	0.02	0.02	0.03	0.12
			t/z	1.49	2.81 *	4.69 *	3.70 *
			p	0.136	0.005	<0.001	<0.001
	Embedded word segmentation	Embedded word plausibility	b	0.03	0.12	0.17	0.28
			SE	0.02	0.02	0.03	0.12
			t/z	1.63	5.05 *	6.71 *	2.32 *
			p	0.103	<0.001	<0.001	0.020
	Incremental word segmentation	Embedded word plausibility	b	0.02	0.14	0.16	0.11
			SE	0.02	0.02	0.03	0.11
			t/z	1.25	5.84 *	6.37 *	0.97
			p	0.212	<0.001	<0.001	0.331

Note. Asterisks indicate statistically significant effects, *p* < 0.05. Converged model for FFD, epvar.lmer = lmer(depvar ~ PI * COLOR * GROUP + (1 + PI|participant) + (1|item), datafile). Converged model for GD, TRT: depvar.lmer = lmer(depvar ~ PI * COLOR * GROUP + (1|participant) + (1|item), datafile). Converged model for RI: depvar.glmer = glmer(depvar ~ PI * COLOR * GROUP + (1|participant) + (1|item), datafile, family = binomial).

## Data Availability

The data in this study are available from https://figshare.com/s/d09c2b3cefc44a65f21c, from 20 June 2025.

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
