# Peer review of "The Impact of Color Cues on Word Segmentation by L2 Chinese Readers: Evidence from Eye Movements"

_behavsci, 2025, doi:10.3390/bs15070904_

Round 1
Reviewer 1 Report
Comments and Suggestions for Authors
This paper tackles a timely issue in Chinese reading, with both theoretical and applied implications, and it does so very competently. I only have a few minor comments.
Lines 90-101. In this review of the literature, it can be indicated that there is published work in unspaced alphabetic writing systems, Javanese, examining the role of alternating color, Wang et al. (2025, 10.1007/s41809-024-00159-1)
Lunes 103-109. There is recent work by Pan et al. (2025, doi:10.3758/s13423-024-02581-6) who found some a small advantage of alternative color segmentation with older Chinese participants,
Table 2. The whole sentences in Chinese should be on the same line. As it stands now, the final character is on the following line of text.
For Table 9, perhaps it is a bit too dense. Not sure whether it’s easy to have it less dense, but the authors may want to have a new look.
Results. In some places comparing two conditions within a parenthesis, I would add “respectively”, for instance on line 594, etc.
For the Discussion, the authors might discuss, for future research, the examination of Overlapping Ambiguous Strings of three characters, middle character can combine with either adjacent character to form real words (see Chen et al., 2025, doi:10.1080/23273798.2024.2390003 for evidence of semantic activation in the absence of context). Actually, I don’t know if this could have been the case in the current study, as it was not mention, but I believe that some of the cited studies had this turf of sentence . Either way, this adds an extra layer of segmentation complexity which would amplify the effects.
For the Conclusions, I prefer to have the five points as continuous text (First… Second…) and perhaps the authors may want to use that option as well—they may keep it as it is, and it’s their paper
The “6.Patent” version should be Author Information (or whichever label is standard in the submission system; it does not appear related to Patent).
Author Response
Reviewer 1:
This paper tackles a timely issue in Chinese reading, with both theoretical and applied implications, and it does so very competently. I only have a few minor comments.
Lines 90-101. In this review of the literature, it can be indicated that there is published work in unspaced alphabetic writing systems, Javanese, examining the role of alternating color, Wang et al. (2025, 10.1007/s41809-024-00159-1)
RESPONSE: We now include reference to this advance publication on L1 reading of Javanese in our Introduction.
Lunes 103-109. There is recent work by Pan et al. (2025, doi:10.3758/s13423-024-02581-6) who found some a small advantage of alternative color segmentation with older Chinese participants,
RESPONSE: We had already cited an advance publication of this article as Pan et al., 2024. We have now updated the reference.
Table 2. The whole sentences in Chinese should be on the same line. As it stands now, the final character is on the following line of text.
RESPONSE: We have adjusted Table 2 to ensure all Chinese sentences remain on the same line, as suggested.
For Table 9, perhaps it is a bit too dense. Not sure whether it’s easy to have it less dense, but the authors may want to have a new look.
RESPONSE: We appreciate the reviewer's suggestion. We have carefully reconsidered the table's structure and would like to explain our design rationale:
- The current version already represents a simplified presentation after multiple iterations of refinement. Table 9 serves as a comprehensive summary of statistical effects that are fundamental to our core analyses.
- The table is systematically organized to facilitate comparison between the two language groups (L1 vs. L2 Chinese speakers), which is the primary focus of our study:
Lines 1: presents the main effect of Language Group
Lines 2-8: show effects for L1 Chinese readers
Lines 9-15: present parallel analyses for L2 Chinese readers
- Within each group, the results follow a logical progression:
Main effect of embedded word plausibility (Lines 2 & 9)
Main effects of color conditions (Lines 3-5 & 10-12)
Plausibility × color interactions (Lines 6-8 & 13-15)
While we acknowledge the table contains substantial information, we believe this organization best presents the complete pattern of results necessary for interpreting our key findings.
Results. In some places comparing two conditions within a parenthesis, I would add “respectively”, for instance on line 594, etc.
RESPONSE: We have added “respectively” in parentheses where two conditions are compared (e.g., line 594) to improve clarity.
For the Discussion, the authors might discuss, for future research, the examination of Overlapping Ambiguous Strings of three characters, middle character can combine with either adjacent character to form real words (see Chen et al., 2025, doi:10.1080/23273798.2024.2390003 for evidence of semantic activation in the absence of context). Actually, I don’t know if this could have been the case in the current study, as it was not mention, but I believe that some of the cited studies had this turf of sentence . Either way, this adds an extra layer of segmentation complexity which would amplify the effects.
RESPONSE: We now note that future research could examine a broader range of segmental ambiguity, including the overlapping ambiguity investigated by Chen et al. (2025). There are certainly similarities between their design and ours. This is unsurprising as the Chen et al. study, like our study, was inspired by the earlier work by Zhou and Li (2021).
For the Conclusions, I prefer to have the five points as continuous text (First… Second…) and perhaps the authors may want to use that option as well—they may keep it as it is, and it’s their paper
RESPONSE: We are grateful to the reviewer for their comments. We have quite complex conclusions and had chosen to enumerate these as discrete points to aid their comprehension. As suggested, we now present our five points as continuous text, using “First…Second… etc.” to delineate the points.
The “6.Patent” version should be Author Information (or whichever label is standard in the submission system; it does not appear related to Patent).
RESPONSE: we have now removed the “6.Patent” heading.
Reviewer 2 Report
Comments and Suggestions for Authors
This study is of great interest for exploring reading processes in Chinese, particularly in the case of bilinguals.
Overall, it is very well written, referenced and justified.
The most important problem I could highlight is related to the sample. The population is not very homogeneous. It would be interesting to be able to increase the number of participants according to their mother tongue in future studies. In this way, it could be studied whether having Russian as a mother tongue as opposed to Japanese (whose writing system shares the kanji, but the words are segmented with their own syllabary) could be a disadvantage in the reading of ambiguities. Or, on the contrary, regardless of the native language, there will be no advantage in this respect.
Author Response
Reviewer 2:
This study is of great interest for exploring reading processes in Chinese, particularly in the case of bilinguals.
Overall, it is very well written, referenced and justified.
The most important problem I could highlight is related to the sample. The population is not very homogeneous. It would be interesting to be able to increase the number of participants according to their mother tongue in future studies. In this way, it could be studied whether having Russian as a mother tongue as opposed to Japanese (whose writing system shares the kanji, but the words are segmented with their own syllabary) could be a disadvantage in the reading of ambiguities. Or, on the contrary, regardless of the native language, there will be no advantage in this respect.
RESPONSE: We agree that such issues could be investigated further using L2 samples with the same or similar native language, and note this in the Discussion.
Reviewer 3 Report
Comments and Suggestions for Authors
The authors examined how Mandarin L1 vs. L2 readers resolve ambiguity as present in compound (incremental?) words that contain an embedded word plus a third element that changes the meaning of the former. Specifically, they wanted to know (1) whether L2 readers have more difficulty than L1 in resolving the situation and segmenting the word correctly and if (2) the L2 group benefitted from color-coding the text. As far as I understand, results indicated that (1) no - there was no evidence that the L2 group had more difficulties in the segmentation task and (2) yes - L2 (but not L1) benefited from color-coding.
This study seems to have been conducted very carefully. The text is well-organized and quite detailed. However, I think that this focus on the details, combined with the complexity of the design and density of information, renders the paper a bit hard to read, in the sense that it is not easy to keep in mind what the big question is about (except in the abstract). Other issues relate to the multiple eye-tracking measures, and also the way results are presented. I present my concerns below with more detail.
1-Keeping the big picture active
It was hard for me to keep in mind
- the distinction between the questions and the assumptions, e.g., is it assumed that readers will use a parallel strategy, or is this also a question to be answered from the results?
- The questions themselves, since it is not always clear that ambiguity resolution is the only focus. The study seems to be addressing general reading efficiency too (from sentence analyses)
- The correspondence between the questions and the experimental manipulations, i.e., what in the manipulations served question 1, question 2, or some other.
I think that a clarification of these issues will improve greatly the experience of reading this interesting paper.
2-Choice and handling of multiple eye-tracking measures (gaze time, etc)
This is a recurrent issue in eye-tracking studies, and it engages multiple questions:
-why were multiple measures necessary?
-what was the motivation for choosing those specific measures?
-How did the authors protect themselves from “cherry-picking” (i.e., collecting several measures while endorsing the presence of supporting evidence from only a few) and/or the risk of inflated significant results by considering multiple measures?
Note that I understand the need for first- vs. second-pass measures, but, beyond this, further justifications seem needed.
3-Presentation of results
The paper contains multiple tables, some of them very long and covering more than one page. I don’t think this table-based approach favours readability, even though the contents are important. I wonder if
-the authors could reallocate some of the table contents at an appendix and invest more on graphic representations.
-the authors could prioritize the visual depiction of the essentials in a simpler way – i.e., focusing on the response to the two main questions announced at the abstract.
Please note that I am not suggesting that the reasoning is simplified. Instead, I am challenging the authors to transform the inherent complexity of this study into a more readable product simply by prioritizing the core message.
Author Response
Reviewer 3:
The authors examined how Mandarin L1 vs. L2 readers resolve ambiguity as present in compound (incremental?) words that contain an embedded word plus a third element that changes the meaning of the former. Specifically, they wanted to know (1) whether L2 readers have more difficulty than L1 in resolving the situation and segmenting the word correctly and if (2) the L2 group benefitted from color-coding the text. As far as I understand, results indicated that (1) no - there was no evidence that the L2 group had more difficulties in the segmentation task and (2) yes - L2 (but not L1) benefited from color-coding.
RESPONSE: Our results are quite complex, which is why we summarize them as five points in the Conclusion. The L2 readers clearly experienced more reading difficulty than the L1 readers and appear to employ a more cautious character-by-character reading strategy. Both groups appear able to process incremental ambiguities, although the L2 readers may do so more slowly. Interestingly, as the reviewer notes, the L2 readers appear to make greater use of visual cues provided by alternating text color.
This study seems to have been conducted very carefully. The text is well-organized and quite detailed. However, I think that this focus on the details, combined with the complexity of the design and density of information, renders the paper a bit hard to read, in the sense that it is not easy to keep in mind what the big question is about (except in the abstract). Other issues relate to the multiple eye-tracking measures, and also the way results are presented. I present my concerns below with more detail.
1-Keeping the big picture active
It was hard for me to keep in mind
- the distinction between the questions and the assumptions, e.g., is it assumed that readers will use a parallel strategy, or is this also a question to be answered from the results?
- The questions themselves, since it is not always clear that ambiguity resolution is the only focus. The study seems to be addressing general reading efficiency too (from sentence analyses)
- The correspondence between the questions and the experimental manipulations, i.e., what in the manipulations served question 1, question 2, or some other.
I think that a clarification of these issues will improve greatly the experience of reading this interesting paper.
RESPONSE: As noted above, we acknowledge that our results are complex. We have further revised the manuscript to clarify the research questions and their connection to the analyses. These revisions include the addition of sub-headings to delineate sections more clearly, as well as a new section at the end of the Introduction that summarizes the research questions. We highlight the major changes in green text but have taken the opportunity to make smaller revisions across the text to enhance its readability.
2-Choice and handling of multiple eye-tracking measures (gaze time, etc)
This is a recurrent issue in eye-tracking studies, and it engages multiple questions:
-why were multiple measures necessary?
-what was the motivation for choosing those specific measures?
-How did the authors protect themselves from “cherry-picking” (i.e., collecting several measures while endorsing the presence of supporting evidence from only a few) and/or the risk of inflated significant results by considering multiple measures?
Note that I understand the need for first- vs. second-pass measures, but, beyond this, further justifications seem needed.
RESPONSE: We agree that this is an important issue in eye movement research. We have tried to interpret our findings conservatively, by focusing on key measures and avoiding cherry-picking. We have now formalized this further in the revision by acknowledging the issues that the reviewer raises (p11-12) and being explicit about the key measures that allow us to conduct hypothesis-driven tests and acknowledging where our analyses are exploratory.
3-Presentation of results
The paper contains multiple tables, some of them very long and covering more than one page. I don’t think this table-based approach favours readability, even though the contents are important. I wonder if
-the authors could reallocate some of the table contents at an appendix and invest more on graphic representations.
-the authors could prioritize the visual depiction of the essentials in a simpler way – i.e., focusing on the response to the two main questions announced at the abstract.
Please note that I am not suggesting that the reasoning is simplified. Instead, I am challenging the authors to transform the inherent complexity of this study into a more readable product simply by prioritizing the core message.
RESPONSE: We have adjusted the format of the tables to make them less disruptive to reading. We have also considered the reviewer’s points carefully. Our view is that the information in these tables is sufficiently important to our presentation of findings for these to remain the main text, although we appreciate the reviewer’s concerns.
Round 2
Reviewer 3 Report
Comments and Suggestions for Authors
The authors have done a good job – especially in justifying eye-tracking measures – but I think there is room for improvement as far as readability is concerned. I respect the authors’ choice for keeping all tables.
1-What I cannot find satisfactory is the prevailing lack of clarity regarding goals vs. findings. To illustrate my point, I selected several excerpts and tried to synthesize each of these as an effect of an independent variable(s) on a dependent variable(s). In my understanding, the paper deals with color and or group as independent variables and reading fluency, ambiguity resolution and parafoveal processing as dependent ones. My point is that the goals (effects to be tested) and conclusions (obtained effects) seem to be changing across the text. Plese note that I am assuming that ambiguity resolution is indexed by (target) word-level analysis. When I write group x color, I am referring to the analysis of both main effects and the interaction.
I hope this helps the authors to put themselves in the reader’s position.
Abstract
“Our study investigated: (1) whether L2 readers experience 12
greater difficulty processing temporary segmental ambiguities compared to L1 readers,
and,,,”
- Group on ambiguity processing
“…(2) whether visual boundary cues can facilitate ambiguity resolution in L2 reading.”
- Color on ambiguity resolution for L2 readers
Discussion
“The present research used eye movement measures to investigate the influence of visual boundary cues on on-line segmental ambiguity processing by skilled native (L1) Chinese readers and high-proficiency second language (L2) readers.”
- Group x color on ambiguity resolution
“This approach allowed us to test several hypotheses concerning L2 Chinese sentence processing relating to (1) overall L1 vs. L2 differences in eye movement behavior,”
- Group on overall reading fluency (ambiguity and all the rest)
“… (2) processing of segmental ambiguity,”
- ambiguity resolution in general? Just in L2?
“ (3) effects of text color-coding on ambiguity processing, “
- color on ambiguity resolution
“…and (3) effects of parafoveal processing. “
- As I outline below, the results suggest that parafoveal processing is a dependent variable. Thus, it is intriguing to see the expression “effects of parafoveal processing”
Conclusion
“First, compared to L1 readers, L2 readers exhibit substantial reading difficulty, em-
ploying a cautious character-by-character reading strategy to achieve comprehension,
even when color-coding marks word boundaries correctly. “
- Group x color on overall reading fluency (ambiguity and all the rest)
“Second, using alternating text-color to mark word boundaries appears to provide lit-
tle overall benefits to reading fluency for either skilled L1 or high-proficiency L2 readers. “
- same as above?
“Third, skilled L1 and high-proficiency L2 readers can process incremental segmental
ambiguities highly effectively, suggesting shared segmentation strategies. “
- group on ambiguity processing
Fourth, L2 readers nevertheless show greater sensitivity to color-based cues when
processing segmental ambiguities, such that cuing a plausible embedded word analysis
appeared to increase competition between this analysis and the alternative whole-word
analysis, inflating reading times for the ambiguity.
- group x color on ambiguity processing
Finally, both L1 and L2 readers appear able to parafoveally process complex semantic
information about upcoming words, enabling them to parafoveally assess the plausibility
of an embedded word analysis prior to fixating a segmental ambiguity. However, this
parafoveal processing appears to be largely unaffected by color cues.
- Group x color on parafoveal preview
2-Please clarify what is the purpose of manipulating and testing the effects of embedded word plausibility. Is it to understand whether there is parallel or serial processing?
Author Response
POINT 1: The authors have done a good job – especially in justifying eye-tracking measures – but I think there is room for improvement as far as readability is concerned. I respect the authors’ choice for keeping all tables.
RESPONSE: We’re grateful to the reviewer for these positive comments.
POINT 2: 1-What I cannot find satisfactory is the prevailing lack of clarity regarding goals vs. findings. To illustrate my point, I selected several excerpts and tried to synthesize each of these as an effect of an independent variable(s) on a dependent variable(s). In my understanding, the paper deals with color and or group as independent variables and reading fluency, ambiguity resolution and parafoveal processing as dependent ones. My point is that the goals (effects to be tested) and conclusions (obtained effects) seem to be changing across the text. Plese note that I am assuming that ambiguity resolution is indexed by (target) word-level analysis. When I write group x color, I am referring to the analysis of both main effects and the interaction.
I hope this helps the authors to put themselves in the reader’s position.
RESPONSE: Our study focuses effects of color cues (mono-color, color coding for an embedded-word analysis, color-coding for a whole-word analysis) on the processing of a segmental ambiguity by L1 and L2 Chinese readers. We examine effects in sentence-level and word-level analyses where we have, guided by the reviewers, carefully delineated those that are hypothesis-testing measures and those that are exploratory. The richness of our data enables us to comment on the influence of both individual factors (L1 vs. L2 readers), effects of color-coding text on reading, segmental ambiguity processing, as well as interactions of these factors, including the key interactions that are the focus of our hypotheses. Moreover, given the opportunity to examine effects at both a region of text containing the ambiguity (the target region) and a prior region that constrains the plausibility of alternative analyses of that ambiguity (the pre-target verb region), we were able to report on both the fixational processing and parafoveally processing of the ambiguity.
We appreciate this creates a rich dataset and rather extensive reporting of findings. We also appreciate the reviewer’s concern that we report this information as clearly as possible. We have therefore further revised extensive sections of the paper to enhance the clarity of our reporting of findings. The reviewer’s highlighting of the need to ensure consistency across different sections of the report (as shown below) was helpful in doing so. We have tried to ensure consistency across sections, better labelled sections of the Discussion to delineate sets of analyses, and revised the manuscript extensively to more clearly signpost the reasons for and implications of the different analyses we conduct. We have also revisited our conclusions to ensure these are more clearly related to the earlier sections. We hope that these extensive revisions clarify our reporting of the findings and makes it easier for readers to understand what we are trying to achieve in this manuscript.
Abstract
“Our study investigated: (1) whether L2 readers experience 12
greater difficulty processing temporary segmental ambiguities compared to L1 readers,
and,,,”
- Group on ambiguity processing
“…(2) whether visual boundary cues can facilitate ambiguity resolution in L2 reading.”
- Color on ambiguity resolution for L2 readers
Discussion
“The present research used eye movement measures to investigate the influence of visual boundary cues on on-line segmental ambiguity processing by skilled native (L1) Chinese readers and high-proficiency second language (L2) readers.”
- Group x color on ambiguity resolution
“This approach allowed us to test several hypotheses concerning L2 Chinese sentence processing relating to (1) overall L1 vs. L2 differences in eye movement behavior,”
- Group on overall reading fluency (ambiguity and all the rest)
“… (2) processing of segmental ambiguity,”
- ambiguity resolution in general? Just in L2?
“ (3) effects of text color-coding on ambiguity processing, “
- color on ambiguity resolution
“…and (3) effects of parafoveal processing. “
- As I outline below, the results suggest that parafoveal processing is a dependent variable. Thus, it is intriguing to see the expression “effects of parafoveal processing”
Conclusion
“First, compared to L1 readers, L2 readers exhibit substantial reading difficulty, em-
ploying a cautious character-by-character reading strategy to achieve comprehension,
even when color-coding marks word boundaries correctly. “
- Group x color on overall reading fluency (ambiguity and all the rest)
“Second, using alternating text-color to mark word boundaries appears to provide lit-
tle overall benefits to reading fluency for either skilled L1 or high-proficiency L2 readers. “
- same as above?
“Third, skilled L1 and high-proficiency L2 readers can process incremental segmental
ambiguities highly effectively, suggesting shared segmentation strategies. “
- group on ambiguity processing
Fourth, L2 readers nevertheless show greater sensitivity to color-based cues when
processing segmental ambiguities, such that cuing a plausible embedded word analysis
appeared to increase competition between this analysis and the alternative whole-word
analysis, inflating reading times for the ambiguity.
- group x color on ambiguity processing
Finally, both L1 and L2 readers appear able to parafoveally process complex semantic
information about upcoming words, enabling them to parafoveally assess the plausibility
of an embedded word analysis prior to fixating a segmental ambiguity. However, this
parafoveal processing appears to be largely unaffected by color cues.
- Group x color on parafoveal preview
2-Please clarify what is the purpose of manipulating and testing the effects of embedded word plausibility. Is it to understand whether there is parallel or serial processing?
RESPONSE: We hope that our revision is now clear that our aim is to determine whether L1 and L2 readers use the same strategies to process the ambiguity and to examine whether color-coding the text influences this processing. To do so we use a plausibility manipulation to determine whether either group of readers adopt the embedded word analysis of the ambiguity, which current theorizing (Li & Pollatsek, 2020) consider to be the dis-preferred analysis. The plausibility manipulation we use is inspired by previous research by Li & Zhou (2021) and also Li et al. (2024). This provides a powerful technique for investigating segmental ambiguity processing.
Our findings show considerable similarity in the processing of the ambiguity by L1 and L2 readers, which we take to be consistent with other research suggesting that L2 readers achieve proficiency in this aspect of sentence processing. We also show limited effects of color coding on the processing of this ambiguity – although these are stronger for the L2 than L1 readers. We also observe intriguing effects at the pre-target region suggesting that both sets of readers appear to adopt an embedded analysis of the ambiguity during parafoveal processing. Moreover, this effect is especially strong for the L2 readers. All indications are that this is short-lived and that the whole-word analysis ultimately is selected by both groups of readers. We consider this effect to be potentially important as it contrasts with the account presented by Li & Pollatsek (2020) and Li et al. (2021).
We have revised parts of the manuscript to further highlight our reason for manipulating plausibility and to make the points we summarize above more clearly in the manuscript. We hope that the reviewer finds our account of these effects to be clearer.